# Public Health Implications of Invasive Plants: A Scientometric Study

**DOI:** 10.3390/plants12030661

**Published:** 2023-02-02

**Authors:** Camila Denóbile, Wagner Antonio Chiba de Castro, Dalva Maria da Silva Matos

**Affiliations:** 1Graduate Program in Neotropical Biodiversity, Federal University of Latin American Integration, UNILA, Foz do Iguaçu 85870-901, Brazil; 2Latin American Institute of Life and Nature Sciences, Federal University of Latin American Integration, UNILA, Foz do Iguaçu 85870-901, Brazil; 3Department of Hydrobiology, Federal University of São Carlos, UFSCar, São Carlos 13565-905, Brazil

**Keywords:** biological invasion, alien plants, facilitation, disease risk, plant interactions, One Health

## Abstract

Movements of organisms through distinct places can change the dynamics of ecological interactions and make the habitat conducive to the spread of diseases. Faced with a cyclical scenario of invasions and threats in a One Health context, we conducted a scientometric study to understand how disturbances in environments with invaded vegetation affect the incidence of parasites and disease prevalence rates. The search was carried out in Web of Science and Scopus databases, with keywords delimited by Boolean operators and based on the PRISMA protocol. Thirty-sixarticles were full-read to clarify the interaction between diseases and invaded areas. The analysis covered publications from 2005 to 2022, with a considerable increase in the last ten years and a significant participation of the USA on the world stage. Trends were found in scientific activities, and we explored how invasive species can indirectly damage health, as higher concentrations of pathogens, vectors, and hosts were related to structurally altered communities. This paper reveals invaded plants threats that enhance disease transmission risks. It is likely that, with frequent growth in the number of introduced species worldwide due to environmental disturbances and human interventions, the negative implications will be intensified in the coming years.

## 1. Introduction

Invasive species may modify ecosystems by altering environmental conditions and ecological processes [1]. To enter these ecosystems, one of the main pathways to species introduction is human-mediated activities, such as trade and tourism—that transport organisms between different ranges around the world [2,3].

Biological invasions, in addition to impacts on ecosystem services and economic damages, represent a threat to public health [4,5]. Shifts in geographic distribution impact how species interact [6], and these dynamics enable pathogens to spread broadly [7]. Altered environments may be associated with rises in disease incidence; thus, introduced populations that disrupt the dispersal of disease’s vectors and hosts may display a threat to human and animal health [8,9].

Despite this, the risk of infectious disease rarely is seen as linked to plant introduction processes [10], especially when contrasted to other invasive species, such as arthropods and mammals—that are the agents directly responsible for pathogen transmission [11]. In general, health implications attributed to alien plants include skin irritation, allergies, and poisoning problems, due to the presence of pollen and toxins [12,13]. Though non-native plants have been highlighted as facilitators—as theyprovide high-quality resources and create a more favorable microclimate for certain organisms [14,15]—alien plants also indirectly allow the occurrence and proliferation of diseases [16]. There is an indispensable need to fill research gaps concerning the impacts of associations between invaded areas and disease risk. Invasion ecology plays a crucial role in protecting landscapes and populations from these threats [17]. This intrinsic connection of biotic and abiotic components represents the One Health approach [18], an initiative focused on integrating the environment, humans, and animals to combat the emergence of infectious diseases and ensure the health maintenance of communities [19]. From this, we can expand our knowledge of events that affect biodiversity and, thus, outline priorities to prevent major changes that impact the natural integrity of ecosystems and health [20].

Given the potential danger of invasive species to animal and human health, this paper highlighted the contribution of alien plants within the scope of integrated health. We conducted a scientometric study that explores the impacts of invaded vegetation areas on the incidence of pathogens, vectors, or hosts, in order to realize how environmentaldisturbances affect disease prevalence rates. To obtain a broader perspective of the scientific knowledge constituted in the last decades, the purposes of this study are: (i) evaluate whether areas invaded by plants offer health risks to animals and humans; (ii) identify the relationship between plant invasion and emerging diseases; and (iii) identify geographic distributions and main biological groups that highlight.

## 2. Results

We found a total of 1579 documents in both databases and added the two literature searches. Initially, Web of Science (WOS) indicated 381 results in the main collection for “human” and 189 for “animal”, while the Scopus (SCO) database resulted in 502 publications for “human” and 507 for “animal”. Using a reference manager, after duplicate elimination and applying selection criteria, they were reduced to 51 (WOS) and 44 (SCO). Incorporated results of both platforms at the end of the screening, replicated references (presented in searches “human” and “animal”) were subtracted again, and, with full-text reading, 36 articles matched our criteria and research objectives (Figure 1). This final result comprised predominantly of papers about invasive plants with problems linked to human and animal health (52.7% of the total), but a significant part of the sample (38.8%) is restricted to investigating potential interactions that cause risks only to human health (Table 1).

The qualitative analysis covered research published in the period of 2005 to 2022, and distribution over time showed that scientific activities did not remain constant through the years. There was a considerable increase in the number of worldwide publications, noticed mainly in the last ten years, and only 13.8% of the total is dated before 2010 (Figure 2). Articles were published in 22 different journals (Table 1, including EcoHealth (*n* = 5), Parasites and Vectors (*n* = 3) and Environmental Entomology (*n* = 3). The journals Acta Tropica, Ecological Applications, Journal of Medical Entomology, Journal of Vector Ecology, PLOS One, and Viruses published two articles each, whereas others appeared only once.

Regarding the research types, there was a strong prevalence of field studies (*n* = 22; 61.1%), and laboratory experiments were the second most common research method (*n* = 7; 19.4%). Literature reviews (*n* = 2) and modeling/remote sensing (*n* = 1) were joined in “other” classification because it was a method slightly used (Figure 2).

To assess the contribution of each geographical region to the study field scenario, Figure 3indicates the geographical distribution of articles comprising the theme of invasive species and human or animal health. There is expressive participation of North America inthis research area development. The United States appears as the countrythatpublished most articles worldwide, owning a considerable number of more than half of the results obtained (61.7%).Although with a very discrete number of publications, Australia (*n* = 3) and South Africa (*n* = 2) were the only countries thatdid not present only one publication in the count. There were no records in Central America.

The most cited invasive plant families were Caprifoliaceae (*n* = 5), Araceae (*n* = 4), and Poaceae (*n* = 4) (Figure 3). *Lonicera mackii* was present in all studies relating to Caprifoliaceae, and genus *Pistia* was covered in most articles on Araceae. Poaceae was the most diverse species mentioned, although none was pointed out more than once.

Another important issue to be measured are the agents and types of ecological relationships involved in disease dissemination (Figure 4). When we evaluated such attributes, the most studied relationships were between hosts and pathogens (30.5%), followed by studies that addressed only vector organisms (22.2%) (Figure 4a). About taxonomic groups, we found a trend in papers that referred to vectors: 38.8% of articles investigated the relationshipbetween plants and mosquitoes (Figure 4b), the most well-known disease transmission vectors. Only one other group was pointed out in the vector category, the ticks, but half of that number was obtained for mosquitoes (19.4%). Viruses are the most well-represented pathogens, adding up to almost half of the total (47.8%), followed by the bacteria group (26%), present in studies that mentioned this category. Disease hosts are the better-distributed set, with the greatest diversity of groups represented by articles, but among them, the majority cited mammalian rodents in this category (45%).

In the articles that specified names of diseases that may be impacted by plant introductions, the top two were Lyme and Malaria; each one was cited four times. Two of the Lyme studies examined *Berberis thunbergii*-invaded areas, and also half of the Malaria studies verified the impact of *Parthenium hysterophorus*. All scientific publications on Lyme disease were published in the United States regions, while those emphasizing Malaria were from the African continent (except one study, which did not describe the locality). Some studies were not clearly restricted to a single pathology but referred to public health problems indirectly—for instance, by addressing “tick-borne diseases” or “mosquitoes-borne diseases”.

## 3. Discussion

We highlight studies that associated plant invasions with disease propagation as a pertinent and relatively new topic—as evidenced by the rise in publications, particularlywithin the last ten years. This paper summarizes the global scientific production regarding health risks by invasive plants. In general, it appears that the effects of invasive plants might not be entirely apparent and such species are not treated with concern once the impacts are often hidden [57].

We found several occurrences of exposure to an invader with some kind of toxicity, either to animals or humans. The main cases mentioned allergic reactions to pollen and contact dermatitis [58,59,60,61]. However, these articles addressing close proximity between animals/humans and invasive plants through direct contact were not kept in the analysis.We chose to exclude these articles from the search since these situations concern individualplant toxicity, and their removal enables better delimitation of the relationship of invasions with ecological aspects in new environments. As well, studies on agricultural pest species that cause damage to host vegetation itself—such as fungi and aphids, or those that discuss the threat of phytopathogens to agricultural biosecurity, in other words, focused on crop damage [62]—were not included. Research approaches in the economic scope are still more frequent—as species control and management to mitigate damage costs [29]—and they seem to retain higher amounts ofinfluencein scientific activities.

As highlighted by Mack and Smith [16], the impacts of non-native plants are not well recognized. Generally, invasive species have changed aspects in the introduced environments that, indirectly, can modifythe dispersal mechanisms of pathogens, vectors, or hosts [41,44]. For example, plants provide shelter or nutrients [23,52,56], which influence the growth or survival of these populations, which then are able toadapt to new attractive habitats offered by invasives [38]. Overall, studies aim to investigate these effects, obtaining significant differencesin spatial distribution or behavior in invaded areas [27,31,48].

A higher concentration of pathogens, vectors, or hosts was associated with structurally altered communities, and the results indicate that plant expansions represent an alert for an increase in disease transmission foci. Adalsteinsson and collaborators [22] highlight that vectors showed elevated rates of infection by pathogens in invaded areas. Providing appropriate sites for oviposition or resources to enhance survival and development of certain life stages [28]—non-native plants may create habitats for vectors and hosts that increase the possibilities of animals’ and humans’ exposure to diseases [25]. To understand the characteristics of invaded areas that promote this process, some studies achieved abundance predictors of certain organisms, such as the availability of leaf litter [21,29,43,63] or detritus accumulation of another plant [30].

The two most cited diseases, Lyme [22,32,40,48] and Malaria [43,44,46,54], exhibit different transmission routes, through tick vectors infected by bacteria- and parasite-contaminated mosquito bite, respectively. It is interesting to mention that, in the United States, zones invaded by *Lonicera maackii* are positively associated with mosquito vector survival and abundance [28,52]; *Rosa multiflora* was investigated by Adalsteinsson and collaborators [21,22] and linked to tick-borne diseases risk, which may suggest that sites with this species concentrate ticks. Data such as these alert to the need for continuous monitoring of the risk factors associated with disease incidence, with special attention in these locations (North America regions and the African continent) and also in other habitats with a high incidence of vectors cited or susceptible to invasion. Once detected, the more vulnerable areas—consequently, the area with a higher risk for disease appearance—increases the possibility of allocating intervention resources [64], mainly because many impacts are hidden or are identified with delay [57].

Our considerationstake into account how humans can both amplifyand suffer the consequences of this process [65,66]. International traffic [67] and dispersions through clothing, animals, or vehicles [68] introduce new species that can act as reservoirs when established in new environments or, rather, turn into sources of “pathogen pollution” [69]. Furthermore, events of pathogen diversification and zoonotic outbreaks can be attributed to landscapes that have been significantly altered by human activity [70,71].

Its common practice in this field to investigate invasive species isolated or emphasize a few species that are already well-known in literature [72], but this limitation interferes the understanding of the whole process. Integrative approaches, as applied by One Health strategies, that consider multiple levels of interaction between organisms and environments—including the role of invasive species—become an important interdisciplinary bridge to detect and predict threats to biodiversity and consequent impacts on health [73,74]. Rather than controlling negative effects, a primary step to contain the dangerous events cascade cited here is to prevent new species from entering natural territories [75]—although the latter requires appropriate management strategies and infrastructure.

We assume that plant introduction can increase another species’ incidence by assisting the resource needs of these organisms in new habitats [63], but the role of invasive plants on the vector and host dynamics is not fully predictable. Considering the need to deepen studies involved in this complex system, there is difficulty in determining with conviction whether these interactions always cause indirect effects on animal and human exposure to the infectious agent.

## 4. Materials and Methods

### 4.1. Literature Search

The investigation of general research trends on a particular knowledge area in the last decades has brought valuable information for comprehending the science’s current state and its progress in terms of production, communication, and quantitative aspects [76]. Scientometry is an interdisciplinary branch that allows us to map and measure scientific knowledge from a sequence of methodological procedures, enabling to draw a profile of trends and gaps within a specific research field. To obtain a scientific production overview, we adopted the PRISMA (Preferred Reporting Items in Systematic Reviews and Meta-Analyses) structure. This protocol presents a checklist and a flowchart guiding the steps recommended for conducting systematic reviews [77]. The research was carried out in Web of Science (www.webofscience.com (accessed on 12 September 2022)) and Scopus (www.scopus.com (accessed on 12 September 2022)) databases, chosen for being multidisciplinary, highly credible online platforms with a large number of indexed scientific journals and publications. An initial literature review guided the definition of the most viable expressions to be used in data collection.

The search strategy applied followed the use of special characters to include word variations (*) to indicate compound terms (“”) and delimit keyword sequences (Table 2). The search was conducted without limitation of date range and with terms in the field “topic”, which filters information provided in the title, abstract, and keywords of articles. After consulting the materials, duplicate publications were excluded; in other words, when they resulted from more than one database, only one reference was considered in the count to avoid repetition in the survey set.

### 4.2. Selection Process

Then, we performed an initial screening evaluating studies through the information contained in the title and abstract—as a preliminary recognition that these results were relevant to the theme (by individual reading). To ensure this, documents were excluded if they did not fit previously established criteria. Were kept in dataset only articles that:explore discussions focused on relationships between invasive vegetation and hosts, vectors or pathogens abundance of diseases affecting humans and/or animals;assess how plant species introduction can become a health threat when animals or human populations are exposed to invaded areas;address risks associated with human/animal health by comparing outcomes in exotic versus native plant populations.

We deleted papers that did not mention invasive plants and were not found the full text. Once we assigned bibliographies matching the search criteria, references were added to the set for analysis. The third step involved obtaining the full file andcautiously reading to assess how works relate exotic plant species to diseases and confirm that all met the inclusion criteria. A final screening was performed with the exclusion of records that looked potentially relevant, but as they presented few details in abstracts, only by full-text examination was it possible to identify that did not fit the research objectives.

### 4.3. Analysis of Obtained Studies

Information was extracted and tabulated by exploring the entire content of qualified materials from a deeper examination throughout the text—selecting the sessions considered important to systematize data processing and investigate relevant topics for the present study. The parameters are defined as: (i) author(s), (ii) publication year, (iii) journal, (iv) geographical location, (v) research area, (vi) study type, (vii) human or animal health, (viii) risk attribute (biological group of organisms involved in interaction/pathology study-associated), and (ix) invasive plant (when cited). Some elements were categorized into previously listed options to facilitate results analysis (Table 3). This characterization of the collected bibliography was used to enable better interpretation during the analysis and construction of graphical results.

## 5. Conclusions

So far, scientific work has been focused on trying to track ecosystem changes as a result of invasive plant establishment. However, this effort is insufficient since there is an urgent need to propose tools and evaluate strategies to manage and prevent the rapid spreading of agents and diseases. The narrowing of this gap may be a focus point to ensure answers for this field of science in the coming years.

Vector and host control is one of the most effective ways to minimize disease spreading rates. At the same time, management of invasive plants is required once it representsanother huge and costly problem—which can be improved by establishing partnerships between organizations worldwide to carry out collaborative actions in searches for more effective strategies. We suggest continuous monitoring at various spatial scales and attempts to employ eradication techniques. New models for research approaches can be useful in estimating area distribution, calculating risks, and prioritizing control strategies. Understanding the costs of biological invasion to animal and human health is an urgent request, given that introductions of alien species are rising worldwide.

## Figures and Tables

**Figure 1 plants-12-00661-f001:**
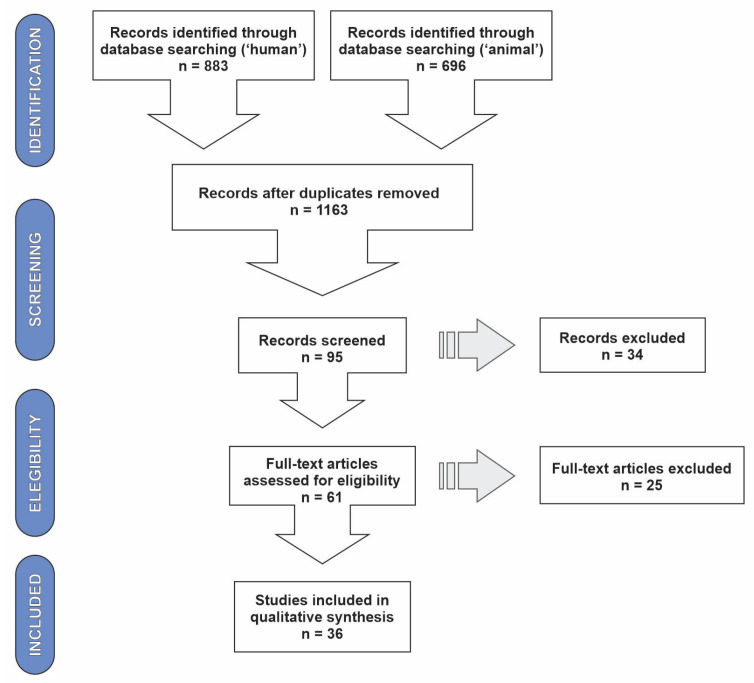
Representation of literature review methodological process elaborated according to PRISMA framework.

**Figure 2 plants-12-00661-f002:**
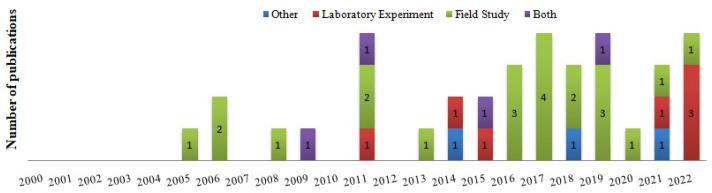
Temporal distribution of studies that related exotic plants with processes that threaten human and/or animal health and proportion of research types.

**Figure 3 plants-12-00661-f003:**
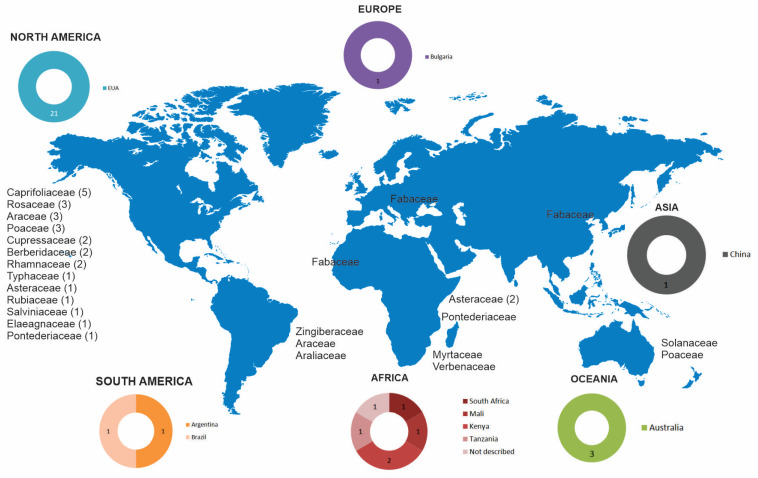
Geographical distribution for each article and invasive plant families, when cited.

**Figure 4 plants-12-00661-f004:**
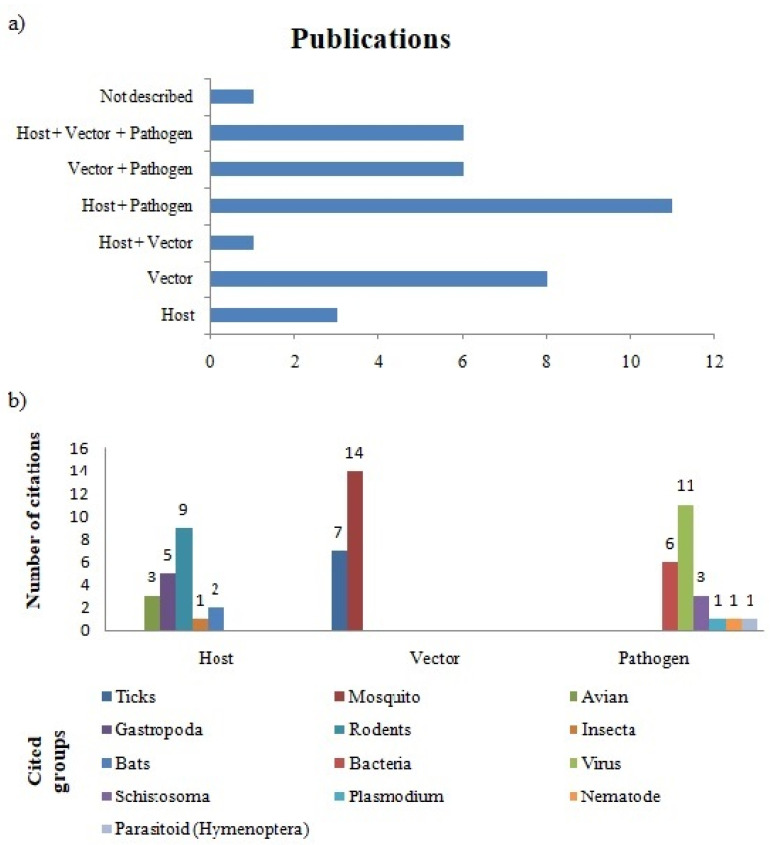
Proportion of biological agents involved in disease dissemination (ordered, in this study, in “risk attributes” parameter) and how they were handled by authors: (**a**) frequency in which each type of interaction between agents leading to the occurrence of disease was evidenced in the dataset; (**b**) taxonomic groups more assigned for each category (host, vector, and pathogen).

**Table 1 plants-12-00661-t001:** List of reviewed articles and their impact target, in health terms.

Reference	Citation	Journal	Impact
[21]	Adalsteinsson et al. (2016)	Ecosphere	Both
[22]	Adalsteinsson et al. (2018)	Parasites & vectors	Both
[23]	Agha et al. (2021)	Viruses	Both
[24]	Andreo et al. (2014)	Viruses	Human
[25]	Blosser et al. (2017)	Acta Tropica	Both
[26]	Buettner et al. (2013)	PLOS One	Animal
[27]	Civitello et al. (2008)	Journal of Medical Entomology	Human
[28]	Conley et al. (2011)	Ecological Applications	Human
[29]	Cuthbert et al. (2019)	Science of the Total Environment	Both
[30]	Desautels et al. (2022)	Acta Tropica	Human
[31]	Desautels et al. (2022)	Hydrobiologia	Human
[32]	Elias et al. (2006)	Journal of Medical Entomology	Both
[33]	Field et al. (2016)	EcoHealth	Both
[34]	Gardner et al. (2015)	Parasites & vectors	Both
[35]	Gardner et al. (2017)	EcoHealth	Both
[36]	Guiden and Orrock (2019)	Behavioral Ecology	Human
[37]	Holsomback et al. (2009)	Journal of Vector Ecology	Human
[38]	Kaestli et al. (2011)	Environmental microbiology	Both
[39]	Leisnham et al. (2019)	International journal of environmental	Both
		research and public health	Both
[40]	Linske et al. (2018)	Environmental Entomology	Both
[41]	Mackay et al. (2016)	Ecological Applications	Both
[42]	Marchetto et al. (2022)	EcoHealth	Animal
[43]	Milugo et al. (2021)	Scientific Reports	Human
[44]	Muller et al. (2017)	Malaria Journal	Human
[45]	Noden et al. (2021)	EcoHealth	Both
[46]	Nyasembe et al. (2015)	PLOS One	Human
[47]	Pearson and Callaway (2006)	Ecology Letters	Human
[48]	Persons and Eason (2019)	Urban Ecosystems	Both
[49]	Plummer (2005)	EcoHealth	Human
[50]	Portman et al. (2011)	Environmental Entomology	Animal
[51]	Reiskind and Zarrabi (2011)	Journal of Vector Ecology	Both
[52]	Shewhart et al. (2014)	Environmental Entomology	Both
[53]	Simeonova et al. (2022)	Current Issues in Molecular Biology	Both
[54]	Stone et al. (2018)	Parasites & vectors	Human
[55]	Teixeira et al. (2017)	Check List	Human
[56]	Wei et al. (2020)	PLOS Neglected Tropical Diseases	Both

**Table 2 plants-12-00661-t002:** Terms combination applied in two separate searches to refine results, both in Web of Science and Scopus platforms.

Search	Keywords Combinations
Animal Health	(“plant invasion*” OR “introduced plant*” OR “plant introduc*” OR “invasive plant*” OR “non-native plant*” OR “exotic plant*” OR “alien plant*” OR “non-indigenous plant*”) AND (“health” OR “disease*” OR “infect*” OR “parasit*” OR “vector*” OR “patho*” OR “host*”) AND (“animal*”)
Human Health	(“plant invasion*” OR “introduced plant*” OR “plant introduc*” OR “invasive plant*” OR “non-native plant*” OR “exotic plant*” OR “alien plant*” OR “non-indigenous plant*”) AND (“health” OR “disease*” OR “infect*” OR “parasit*” OR “vector*” OR “patho*” OR “host*”) AND (“human*”)

**Table 3 plants-12-00661-t003:** Characterization of data collected and selected for qualitative analysis.

Parameters	Categories
Publication Details	Authors
	Title
	Journal
Year	Publication date
Geographic Location	Europe
	Asia
	Africa
	Oceania
	North America
	Central America
	South America
Research Type	Review
	Field Study
	Laboratory Experiment
	Other
Impact	Animal Health
	Human Health
Risk atribute	Disease
	Pathogen
	Vector
	Host
Invasive Plant	Specie
	Family

## Data Availability

Data used in this study can be available by contacting at camiladenobile@gmail.com.

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
