# Peer review of "Public Health Implications of Invasive Plants: A Scientometric Study"

_plants, 2023, doi:10.3390/plants12030661_

Round 1

Reviewer 1 Report

This article presented Public health implications of invasive plants by a scientometric techniques. The study will be useful for evaluation among invasive plants, diseases and invaded areas. Before recommending this article for publication, there are some shortcomings for that should be resolve.

Add a few lines on invasion of plants and its impacts on public health.

Line 38 should be cited with recent study DOI: https://doi.org/10.13133/2239-3129/16869,

Provide links of the soft wares in the methodology.

Also add subsections in the methodology.

Results discussion are well presented.

Add future recommendations and strategies to control invasiveness of species in the conclusion.

Author Response

Thank you for your suggestions. They all were included in the revised version of the manuscript.

  • This article presented Public health implications of invasive plants by a scientometric techniques. The study will be useful for evaluation among invasive plants, diseases and invaded areas. Before recommending this article for publication, there are some shortcomings for that should be resolve.

The authors would like to thank the reviewer for its valuable comments and suggestions which helped, in our opinion, to considerably improve the manuscript.

  • Add a few lines on invasion of plants and its impacts on public health.

A short comment, including references, was added in the revised version, providing further information about plant invasion and its impact on public health.

  • Line 38 should be cited with recent study DOI: https://doi.org/10.13133/2239-3129/16869

We thank you for the suggestion. However, based on the search methodology used which filters information provided in title, abstract and keywords of articles, this paper was not selected. Although some of the plants cited along it are allergenic causing bronchitis and asthma or even narcotic, unfortunately we understand that we may keep out results based on the previously defined systematic search.

  • Provide links of the softwares in the methodology.

We added them in the methodology.

  • Also add subsections in the methodology.

We added them in the methodology.

  • Results discussion are well presented.

We thank you for you comment. You clearly understood the main focus of our work and saw far beyond the simplicity of the method and contribution of these results for further studies.

  • Add future recommendations and strategies to control invasiveness of species in the conclusion.

Thank you again for this suggestion. We add recommendations and call attention for the need of future studies and strategies in the conclusion.

Reviewer 2 Report

The authors use a literature review and synthesis (which they refer to as scientometric study) to determine if there is a connection between plant invasions and disease incidence in a OneHealth context. As I understand it, the basic idea is that plant invasions may create habitat for disease vectors or hosts, and therefore indirectly increase disease incidence. I believe this is an idea that would have broad interest.

Indeed, it does seem that the authors find some support for the idea that plant invasions can be associated with increased disease incidence, although I believe that they recognize that it isn’t entirely clear how commonly this is the case.

My biggest criticism of the paper is that the writing in English was very poor, which made it very difficult to comprehend and evaluate. Specifically, I had a difficult time following some of the objectives, and much of the discussion. In my review, I tried to determine whether the objectives were met, and what the results and synthesis from each was, which was very difficult to do. The stated objectives (purposes) of the paper were:

1)      evaluate and quantify the current development scenario;

2)      recognize the main patterns and gaps in available literature about this scientific field;

3)      identify geographic distributions and main biological groups that highlight;

4)      evaluate the relationship between plant invasion and emerging diseases.

I do not understand what “current development scenario” is in the first objective. Do the authors simply mean how many papers have been written on the subject?

I do not understand what is meant by the wording in objective 2.

I do think that the authors accomplish objective 3 (see Figure 3, and perhaps Table 3 and Figure 2 are also relevant).

I do think that the authors evaluate the relationship between plant invasion and emerging diseases in objective 4. At least, they identify some cases where there seems to be a relationship. However, it seems that perhaps what they are saying is that this is a young field without a lot of papers, and so a detailed understanding of any relationship is still unclear. In other words, there are cases, but we don’t know how common, and what factors would affect whether a plant invasion has an effect on disease incidence or not.

In addition, I believe the Introduction could use some more development. Specifically, for a broad audience, I believe it is important to briefly define or describe what OneHealth is.

I think the paper would benefit from organizing the Results and Discussion specifically around the objectives, so that there is a clearer linkage between the data collection, findings, and objectives.

The paragraph beginning around line 193 was vague, and I found it difficult to understand what the authors meat by most studied relationships were between hosts and pathogens, or only vector organisms. I would also separate the paragraph abut taxonomic groups, because that appears to be a different idea.

I didn’t follow many of the paragraphs in the Discussion. I couldn’t follow the second paragraph at all. The fourth, fifth, and sixth paragraphs were also difficult for me to follow. I believe this is where the authors are trying to say that they found a relationship, but I couldn’t tell. I think the problem was with the writing in English, but I also suspect that there were some organizational issues with those paragraphs, too. I apologize that I cannot be more detailed in my criticism of these paragraphs, but that is because they were difficult to comprehend and follow.

In addition, part of the beginning of the Discussion just describes the literature search, which is more appropriate for the Methods. I don’t think there is anything particularly novel in the literature search and synthesis approach (although it is well described) that warrants featuring it in the Discussion.

Author Response

Thank you for your suggestions. They all were included in the revised version of the manuscript.

  • The authors use a literature review and synthesis (which they refer to as scientometric study) to determine if there is a connection between plant invasions and disease incidence in a OneHealth context. As I understand it, the basic idea is that plant invasions may create habitat for disease vectors or hosts, and therefore indirectly increase disease incidence. I believe this is an idea that would have broad interest.

Exactly. We are glad that it was possible to understand the main ideas of our work and the questions that we sought to explore.

  • Indeed, it does seem that the authors find some support for the idea that plant invasions can be associated with increased disease incidence, although I believe that they recognize that it isn’t entirely clear how commonly this is the case.

Although we tried to answer several doubts about the subject, hope that more explanations will be found in further studies, particularly on points highlighted in this paper as research gaps

3- My biggest criticism of the paper is that the writing in English was very poor, which made it very difficult to comprehend and evaluate. Specifically, I had a difficult time following some of the objectives, and much of the discussion. In my review, I tried to determine whether the objectives were met, and what the results and synthesis from each was, which was very difficult to do. The stated objectives (purposes) of the paper were:

1)      evaluate and quantify the current development scenario;

2)      recognize the main patterns and gaps in available literature about this scientific field;

3)      identify geographic distributions and main biological groups that highlight;

4)      evaluate the relationship between plant invasion and emerging diseases.

I do not understand what “current development scenario” is in the first objective. Do the authors simply mean how many papers have been written on the subject?

I do not understand what is meant by the wording in objective 2.

I do think that the authors accomplish objective 3 (see Figure 3, and perhaps Table 3 and Figure 2 are also relevant).

I do think that the authors evaluate the relationship between plant invasion and emerging diseases in objective 4. At least, they identify some cases where there seems to be a relationship. However, it seems that perhaps what they are saying is that this is a young field without a lot of papers, and so a detailed understanding of any relationship is still unclear. In other words, there are cases, but we don’t know how common, and what factors would affect whether a plant invasion has an effect on disease incidence or not.

We would like to apologize for paragraphs that were confusing. Our challenge in joining the large amount of information may have made understanding difficult. We also agree.

The authors noted the typing and semantics errors and hope that the revised version has been clearer.

The objectives were reevaluated and rewritten in order to connect with the topics of discussion.

4- In addition, I believe the Introduction could use some more development. Specifically, for a broad audience, I believe it is important to briefly define or describe what OneHealth is.

At the introduction we improved the concepts and connected them in a way to address all the main contexts in which our theme are inserted. We kept the most relevant references to justify our ideas.

5- I think the paper would benefit from organizing the Results and Discussion specifically around the objectives, so that there is a clearer linkage between the data collection, findings, and objectives.

We believe that the paper has improved considerably after corrections made following the reviewer’s suggestions.

6- The paragraph beginning around line 193 was vague, and I found it difficult to understand what the authors meat by most studied relationships were between hosts and pathogens, or only vector organisms. I would also separate the paragraph abut taxonomic groups, because that appears to be a different idea.

The paragraph was reformulated.

7- I didn’t follow many of the paragraphs in the Discussion. I couldn’t follow the second paragraph at all. The fourth, fifth, and sixth paragraphs were also difficult for me to follow. I believe this is where the authors are trying to say that they found a relationship, but I couldn’t tell. I think the problem was with the writing in English, but I also suspect that there were some organizational issues with those paragraphs, too. I apologize that I cannot be more detailed in my criticism of these paragraphs, but that is because they were difficult to comprehend and follow.

We also agree and this section was revised and rewritten with the improvements.

8- In addition, part of the beginning of the Discussion just describes the literature search, which is more appropriate for the Methods. I don’t think there is anything particularly novel in the literature search and synthesis approach (although it is well described) that warrants featuring it in the Discussion.

The suggested shifts were done in the revised version.

Reviewer 3 Report

The suitability of the contents of the rows 85-89 for the method section might be questioned

It is not the best decision (just simplification of the work) to delete articles written in other languges. For invasion topic in special, there are lots of excellent studies published in native to the authors languages. At least brief characteristic of the numbers of such publications, species analysed in these articles, countries examined should be summarised in the table or the figure  

 Row 42 – specie ?

I would suggest removel of the paragraph about „grey literature“,  it is strange that reviewers do not know, that monographs or dissertations are reviewed and based on published „non-grey“ papers. And in addition, it is big surprise, that monographs and dissertations theses are placed together with the abstracts

Again – such  data about disease topic in the monographs and dissertations should be briefly stated in the table or figure.

Table 2. „Characterization of data collected and selected for qualitative analysis.“ Logistics is missing and such an analysis is quantification

Table 3 would provide wider information in case journal publishers would be added in the separate column

Row 180: We also seek to understand the contribution of each geographical region to study field scenario.

Families listed are not of bigger value, it is shaped and missleading information. Only list of invasive plant species which are related to threaths posed to animals and human

One group of the researchers publishes once and the other groups are discussing the same study in several publications replicating it in other words. Explanation should be provided for unification of information.

Row 180: We also seek to understand the contribution of each geographical region to study field scenario. In what data obtained it has got reflectance? How to combine such a statement with the figure 3?

Meaning of the circles in the Fig. 3 should be explained and it is absolutely unclear why in the Europe circle Bulgaria is mentioned, the same questions arise looking to the circles of the other continents. What does it mean?

Discussions and conclusions are empty statements. No concluded information concerning the most oftenly analysed invasive plant species as specified disease vectors in one or the other continent. The outcome of reading of the paper is poor: before the reading known fact about possible interaction between invasive plants and diseases of animals and human does not obtain more clear and wider view after introduction to the paper.      

Despite the list in the Table 3, for the reader it would be better (in terms of time less consuming) to have reference list with somehow underlined/labeled 36 references analysed by the authors of the paper under review.

Why allergy topic is missing?

The authors should state what new invasion-diseases information (it does not belong to technical aspect of the collection of literature) is gained in the broad search of references,  compared to some former reviews such as 13 (in special), 15, 16, 25 53, 66, 83

Author Response

Thank you for your suggestions. They all were included in the revised version of the manuscript.

  • The suitability of the contents of the rows 85-89 for the method section might be questioned

 The sentence was reformulated.

  • It is not the best decision (just simplification of the work) to delete articles written in other languages. For invasion topic in special, there are lots of excellent studies published in native to the authors languages. At least brief characteristic of the numbers of such publications, species analysed in these articles, countries examined should be summarised in the table or the figure 

We agree with the referee that it’d be much better to include paper in other languages as well. However, in the systematic review using Web of Science and Scopus databases, we must use keywords combinations. Unless papers in other languages rather than in English provided these keywords in their title, abstract and keywords, they would not be selected, including in our native language. So, we believe this methodology avoid bias in the results.

  • Row 42 – specie ?

The word was rewritten.

  • I would suggest removel of the paragraph about „grey literature“, it is strange that reviewers do not know, that monographs or dissertations are reviewed and based on published „non-grey“ papers. And in addition, it is big surprise, that monographs and dissertations theses are placed together with the abstracts

 The sentence was removed.

  • Again – such data about disease topic in the monographs and dissertations should be briefly stated in the table or figure.

We also agree with the referee that it’d be much better to include paper in other languages as well. However, to avoid bias in the results and follow the PRISMA methodology already used in other scientometric studies we did not include monographs and dissertations at this stage.

  • Table 2. „Characterization of data collected and selected for qualitative analysis.“ Logistics is missing and such an analysis is quantification

 Additional information was included to clarify this point.

  • Table 3 would provide wider information in case journal publishers would be added in the separate column

The name of Journals is provided in the Table 1. Unfortunately, we don’t see why to separate the Journals in another table.

  • Row 180: We also seek to understand the contribution of each geographical region to study field scenario.

Based on this information we may have a general idea about where the problem regarding plant invasions and animal and human health have received more attention and where we still have large gaps. Besides, future analysis should evaluate if there is a relation between the number of papers focusing on these relation and number of studies on invasive plants and number of health problem detected. Considering the current and future scenario of human proximity and impacts in the ecosystem, these information could give some directions about the regions where these problems are sub-notified.

  • Families listed are not of bigger value, it is shaped and missleading information. Only list of invasive plant species which are related to threaths posed to animals and human

We thank you for the suggestion. However, considering the results founded, we understand that it would be interesting to highlight families that most frequently appeared in the selected articles, mainly as a warning such as possible threats that may cause considerable problems in future. We also discussed the impacts of some species along the text.

  • One group of the researchers publishes once and the other groups are discussing the same study in several publications replicating it in other words. Explanation should be provided for unification of information.

One group of authors decided to report all the records recorded while other authors presented results based on a systematic search to avoid bias in the interpretation of results.

We agree with the reviewer that it would be better to link the information from publications, we consider doing this in the discussion section.

  • Row 180: We also seek to understand the contribution of each geographical region to study field scenario. In what data obtained it has got reflectance? How to combine such a statement with the figure 3?

Please see comment 8

  • Meaning of the circles in the Fig. 3 should be explained and it is absolutely unclear why in the Europe circle Bulgaria is mentioned, the same questions arise looking to the circles of the other continents. What does it mean?

The circles in this figure 3 represent the total number of publications per continent, within them separated by countries (legend on the right) and in the text we cited these numbers referred in figure. Bulgaria was the only country in the European continent with one article found in this  search.

  • Discussions and conclusions are empty statements. No concluded information concerning the most oftenly analysed invasive plant species as specified disease vectors in one or the other continent.

Both discussion and conclusions were reformulated. Thank you for your comment.

14- The outcome of reading of the paper is poor: before the reading known fact about possible interaction between invasive plants and diseases of animals and human does not obtain more clear and wider view after introduction to the paper.      

We thank you very much for the valuable comments provided. The paragraphs were reformulated following the considerations and we remain at your disposal for any further questions you may have.

15- Despite the list in the Table 3, for the reader it would be better (in terms of time less consuming) to have reference list with somehow underlined/labeled 36 references analysed by the authors of the paper under review.

In the revised version, the 36 references analyzed by authors are listed in table 1.

16- Why allergy topic is missing?

The authors understand that health implications such as skin irritation, allergies and poisoning problems are due to direct contact with plants (presence of pollen and toxins). In this work, we intend to address interactions between invasive plants and other organisms that, in indirect ways, enables the occurrence and proliferation of diseases.

17- The authors should state what new invasion-diseases information (it does not belong to technical aspect of the collection of literature) is gained in the broad search of references,  compared to some former reviews such as 13 (in special), 15, 16, 25 53, 66, 83

Thank you  again for your comments. In the discussion we supported our arguments from other references also, so we believe that this suggestion has been incorporated into the revised text.

Round 2

Reviewer 3 Report

The first impression, after reading such an article title („Public health implications of invasive plants: a scientometric study“) , is the idea that the article primarily discuss the effects of invasive pollen on humans, since this was the most common topic studied by scientists in invasion biology (topic which caused a lot of doctors attention) and well-known for big significant part of community. The given name is too broad for the substance under consideration and should be narrowed, to reflect its novelty in some way, noting that the review discusses the effects of transmission through viruses and bacteria caused diseases, or something similar to it.

212 row

„articles addressing close proximity between animals/humans and invasive plants through direct contact were not kept in analysis“  (expression is very unlogistic, because it does not separate pollens from viruses and bacterias)  

Like in the former version there was subdivision of literature to grey and unknown color, present version also has some ethic issues:  

17-18 row The analysis covered publications from 2005 to 2022, with considerable increase in the last ten years and a significant participation of USA on the world stage.

Question rises: Is anyone article from USA is top science and reminder country production is already of secondary level?